# RETRACTED: Resveratrol Inhibits the Migration and Metastasis of MDA-MB-231 Human Breast Cancer by Reversing TGF-β1-Induced Epithelial-Mesenchymal Transition

**DOI:** 10.3390/molecules24061131

**Published:** 2019-03-21

**Authors:** Yang Sun, Qian-Mei Zhou, Yi-Yu Lu, Hui Zhang, Qi-Long Chen, Ming Zhao, Shi-Bing Su

**Affiliations:** 1Research Center for Traditional Chinese Medicine Complexity System, Shanghai University of Traditional Chinese Medicine, Shanghai 201203, China; bssunyang@126.com (Y.S.); tazhou@163.com (Q.-M.Z.); ava0048@163.com (Y.-Y.L.); zhanghuiman@126.com (H.Z.); cqlw1975@126.com (Q.-L.C.); mingz@hotmail.com (M.Z.); 2AntiCancer Inc., San Diego, CA 92100, USA

**Keywords:** resveratrol, breast cancer, MDA-MB-231 cell, metastasis, migration, EMT

## Abstract

Metastasis is a major cause of death in patients with breast cancer. In the process of cancer development, epithelial-mesenchymal transition (EMT) is crucial to promoting the invasion and migration of tumor cells. In a previous study, the role of resveratrol in migration and metastasis was investigated in MDA-MB-231 (MDA231) human breast cancer cells and a xenograft-bearing mouse model. Additionally, the related mechanism was explored. In the present study, in vitro Transwell assays showed that resveratrol can inhibit the migration of transforming growth factor (TGF)-β1-induced MDA231 cells in a concentration-dependent manner. An enzyme-linked immunosorbent assay (ELISA) showed that resveratrol can reduce the secretion of matrix metalloproteinase (MMP)-2 and MMP-9. Immunofluorescence was performed to confirm the expression of EMT-related markers. Immunofluorescence assays confirmed that resveratrol changed the expression of the EMT-related markers E-cadherin and vimentin. Western blot analysis demonstrated that resveratrol decreased the expression levels of MMP-2, MMP-9, Fibronectin, α-SMA, P-PI3K, P-AKT, Smad2, Smad3, P-Smad2, P-Smad3, vimentin, Snail1, and Slug, as well as increased the expression levels of E-cadherin in MDA231 cells. In vivo, resveratrol inhibited lung metastasis in a mouse model bearing MDA231 human breast cancer xenografts without marked changes in body weight or liver and kidney function. These results indicate that resveratrol inhibits the migration of MDA231 cells by reversing TGF-β1-induced EMT and inhibits the lung metastasis of MDA231 human breast cancer in a xenograft-bearing mouse model.

## 1. Introduction

One of the most common cancers in women is breast cancer, which seriously impairs the patient’s physical and mental health. Systemic metastasis is the main lethal factor in patients with advanced breast cancer [1]. Triple-negative breast cancer (TNBC) is a special type of breast cancer, which is different from HER2 amplification or hormone receptor-positive breast cancer and has higher tumor invasiveness, higher recurrence rates, and a worse prognosis. As a refractory disease, there is a lack of effective drugs for treating TNBC clinically [2].

Epithelial-mesenchymal transition (EMT) is a process where under some specific physiological or pathological conditions, epithelial cells lose their polarity and become mesenchymal cells with the ability to move freely and actively in the cell matrix [3]. EMT is also characterized by the loss of epithelial cell polarity and the acquisition of stromal features, including (1) decreased expression of cell adhesion molecules, which leads to intercellular junction disruption and cell movement and (2) transformation of the keratin cytoskeleton into a waveform protein cytoskeleton. The rearranged cytoskeleton enhances cell-matrix adhesion, changing the phenotype of the cells and enhancing the ability of the cells to move [4]. In recent years, studies have found that in all types of cells, the EMT phenomenon is closely associated with tumor invasion and metastasis and is quite important in the invasion and metastasis of ovarian cancer, breast cancer, colon cancer, lung cancer, prostate cancer, oral cancer, liver cancer, and various other types of cancer [5].

Studies have also shown that growth factors induce cellular morphological changes, such as epithelial marker expression loss and increased mesenchymal marker expression, which are features of the EMT phenotype. These growth factors include transforming growth factor (TGF)-β, epidermal growth factor (EGF), vascular endothelial growth factor (VEGF), platelet derived growth factor (PDGF), and human growth factor (HGF) [6]. Commonly activated EMT signaling pathways are the TGF-β/Smad, phosphatidylinositide-3 kinase (PI3K)/AKT, MEK/ERK, and WNT/β catenin pathways [7]. In these signaling pathways, the growth factors regulate the expression of proteins such as Twist1, SIP1, Snail1, and Slug (Snail2) [8].

The TGF-β signaling pathway is a widely recognized signaling pathway that mediates and regulates the EMT process. TGF-β is expressed highly in some tumors and affects tumor cells through autocrine and paracrine processes, which consequently induce and maintain EMT [9]. Numerous studies have confirmed that TGF-β in liver cancer cells can induce the occurrence of EMT [10]. TGF-β can also promote the expression of matrix metalloproteinase-2/9 (MMP-2/9), which participate in membrane degradation [11].

Resveratrol (trans-3,4,5-trihydroxystilbene) is a type of polyphenol. It is a natural compound that exists in more than 70 types of plants, such as hellebore, giant knotweed, grapes, and peanuts [12]. Resveratrol can regulate human lipid metabolism, inhibit platelet aggregation, protect cardiovascular tissue, and have anti-inflammatory and antioxidant effects. This compound has many types of biological activities and pharmacological effects [13]. Studies have shown that resveratrol can inhibit the initiation and progression of various tumors, such as liver cancer and breast cancer [14,15]. Studies have confirmed that in lung cancer cells, resveratrol can inhibit TGF-β1-induced EMT and suppress the invasion and metastasis processes of lung cancer cells [16]. A large number of preclinical studies have confirmed that resveratrol has an inhibitory effect on breast cancer in vivo and in vitro, and its mechanism is related to various molecular targets such as cell proliferation, EMT/metastasis, apoptosis, epigenetic response, and increased sensitivity to chemotherapy [17]. Resveratrol can also inhibit the invasion and migration of human breast cancer MCF-7 cells through the PI3K/Akt and Wnt/β-catenin signaling pathways [18]. However, no study has explained whether resveratrol can reverse TGF-β1-induced EMT and inhibit the migration of MDA-MB-231 (MDA231) breast cancer cells.

In the present study, the effects of resveratrol on TGF-β1-induced EMT in TNBC cells (MDA231) were investigated. Our findings suggest that resveratrol inhibits the migration and metastasis of MDA231 human breast cancer cells by reversing TGF-β1-induced EMT.

## 2. Results

### 2.1. Resveratrol Inhibits the Migration of Breast Cancer Cells

MDA231, MDA-MB-453 (MDA453), MDA-MB-436 (MDA436), and BT549 (BT-549) cells were treated with different concentrations of resveratrol for three days to determine the effect of resveratrol on breast cancer cell survival. An 3-(4,5-dimethylthiazol-2-yl)-5-(3-carboxymethoxyphenyl)-2-(4-sulfophenyl)-2H-tetrazolium (MTS) assay showed that the survival of cells treated with resveratrol at concentrations of 12.5, 25, 50, and 100 μM for 48 h and 72 h was decreased significantly (compared to untreated cells, *p* < 0.05) (Figure 1). Because cell proliferation was not affected in the MTS experiment, we selected resveratrol concentrations of 12.5, 25, and 50 μM and observed their effect on breast cancer cell migration using Transwell migration assays (Figure 2). Resveratrol inhibited the migration of MDA231 cells at concentrations of 12.5, 25, and 50 μM, and the degree of cell migration inhibition was concentration-dependent. However, resveratrol did not significantly inhibit the migration of MDA436 cells (Figure 2). Resveratrol at concentrations of 25 and 50 μM inhibited BT549 cell migration, with migration rates of 70% and 65%. When the concentration of resveratrol reached 50 μM, MDA453 cell migration was inhibited, and the migration rate was 63% (Figure 2). These results suggest that resveratrol inhibits the migration of breast cancer cells, particularly MDA231 cells.

### 2.2. Resveratrol Reverses TGF-β1-Induced EMT in MDA231 Cells

EMT was previously induced in MDA231 cells by TGF-β1 [19,20,21] to determine whether resveratrol could inhibit EMT. MDA231 cells were treated with 5 ng/mL TGF-β1 or 5 ng/mL TGF-β1 combined with 50 μM resveratrol for 24 h. After treatment with TGF-β1, the MDA231 cells exhibited a scattering phenomenon and lost the intercellular connections observed in the untreated group (Figure 3A). After treatment with TGF-β1 combined with 50 μM resveratrol for 24 h, the morphology of MDA231 breast cancer cells was similar to the morphology of the untreated group (Figure 3A).

To explore the potential role of resveratrol in breast cancer metastasis by inhibiting TGF-β1-induced EMT, we evaluated the effects of resveratrol on migration using Transwell migration assays in vitro. To verify whether resveratrol inhibits cell migration, MDA231 cells were treated with 5 ng/mL TGF-β1 and 0, 12.5, or 50 µM resveratrol; then, the cells were allowed to migrate in Transwell chambers for 24 h. The number of migrated cells was reduced by resveratrol in a dose-dependent manner (*p* < 0.05 or *p* < 0.01; Figure 3B).

To further confirm whether resveratrol can inhibit TGF-β1-induced EMT, MDA231 cells were treated with 5 ng/mL TGF-β1 (TGF-β1 group), 5 ng/mL TGF-β1 + 12.5 μM resveratrol (T + R 12.5 group), 5 ng/mL TGF-β1 + 25 μM resveratrol (T + R25 group), or 5 ng/mL TGF-β1 + 50 μM resveratrol (T + R50 group) for 24 h to examine changes in EMT markers. As shown in Figure 3C, the Western blot analysis showed that the expression of E-cadherin was significantly increased and the expression levels of fibronectin, vimentin, Snail1, Slug, and α-SMA were significantly decreased by resveratrol treatment (*p* < 0.05 or *p* < 0.01). Compared with the control group, TGF-β1 significantly downregulated E-cadherin expression (*p* < 0.05) and upregulated fibronectin, vimentin, Snail1, Slug, and α-SMA expression (*p* < 0.05; Figure 3C).

We also verified the expression of EMT markers by immunofluorescence. TGF-β1 decreased the expression of E-cadherin and increased the expression of vimentin; these effects were reversed by 50 μM resveratrol (Figure 3D). These studies showed that resveratrol reverses TGF-β1-induced EMT in MDA231 cells through upregulating E-cadherin and downregulating vimentin.

### 2.3. PI3K/AKT and Smad Signaling Mediates EMT in TGF-β1-Induced MDA231 Cells

The above data showed that resveratrol could inhibit the migration of TGF-β1-induced MDA231 cells and influence interstitial cell morphology. Additionally, resveratrol can reverse the expression of TGF-β1-induced EMT markers. To further confirm the pathways related to EMT that were induced by TGF-β1 in MDA231 cells, the PI3K inhibitor LY290042, the Smad inhibitor SB431542, the JunN-terminalkinase (JNK) inhibitor SP600125, the NF-κB inhibitor PDTC, and the ERK inhibitor PD98059 were used in conjunction with TGF-β1. The results showed that compared with the TGF-β1 group, the cells treated with TGF-β1 combined with SB431542 or LY294002 expressed high levels of E-cadherin and low levels of vimentin, fibronectin, Snail1, and Slug (*p* < 0.05 Figure 4A). The levels of the EMT markers were not significantly altered by treatment with the JNK inhibitor SP600125, the NF-κB inhibitor pyrrolidinedithiocarbamate ammonium (PDTC), or the ERK inhibitor PD98059 (*p* > 0.05 Figure 4A). When LY290042 or SB431542 were combined with TGF-β1, MDA231 cells regained the epithelial phenotype, which consisted of tight connections between cells, and the cell morphology was consistent with that of the control group (Figure 4B). To verify whether the LY290042 and SB431542 inhibitors can inhibit TGF-β1-induced cell migration, MDA231 cells were treated with 5 ng/mL TGF-β1, 5 ng/mL TGF-β1 + 10 μM LY290042, or 5 ng/mL TGF-β1 + 10 μM SB431542. Cell migration in Transwell chambers was observed for 24 h. LY290042 and SB431542 reduced the TGF-β1-induced migration of cells (*p* < 0.05 Figure 4C). These results suggest that TGF-β1 may induce EMT in MDA231 cells via the PI3K/AKT and Smad signaling pathways.

### 2.4. Resveratrol Reverses TGF-β1-Induced EMT Through the PI3K/AKT, Smad, and MMP Signaling Pathways

To determine whether the EMT-related proteins were affected by resveratrol, the levels of MMP-2, MMP-9, PI3K, P-PI3K, AKT, P-AKT, Smad2, Smad3, P-Smad2, and P-Smad3 were evaluated in MDA231 cells that were treated with or without TGF-β1 and resveratrol. Western blotting analyses showed that levels of P-PI3K, P-AKT, MMP-2, MMP-9, Smad2, Smad3, P-Smad2, and P-Smad3 were decreased by resveratrol treatment (*p* < 0.05; Figure 5A). The levels of PI3K and AKT were not significantly altered by resveratrol treatment (*p* > 0.05; Figure 5A). These results suggest that resveratrol may inhibit MDA231 cell migration and EMT by regulating PI3K/AKT and Smad signaling.

MMP-2 and MMP-9 play important roles in the metastasis of cancer by facilitating extracellular matrix (ECM) degradation. The protein expression levels and secretion of MMP-9 and MMP-2 in MDA231 cells were detected after resveratrol treatment. As shown in Figure 5A, resveratrol effectively reduced the protein expression levels of MMP-2 and MMP-9 in MDA231 cells. Resveratrol also reduced the secretion of MMP-9 and MMP-2 from cells (*p* < 0.01; Figure 5B).

### 2.5. Effects of Resveratrol on MDA231 Xenografts

Since previous studies have shown that resveratrol can effectively prevent invasion in vitro, we investigated its ability to inhibit metastasis in vivo. An athymic nude mouse model of breast cancer was treated with 40 mg/kg resveratrol and vehicle (sham-treated) for two months. Then, the lungs were excised from the sacrificed nude mice and stained with H and E for the histological examination of the number of metastatic nodules (Figure 6A). On average, 20 tumor nodules were detected in the sham-treated group, while 12 nodules were found in the resveratrol-treated group (Figure 6B). These results showed that resveratrol can effectively inhibit the lung transplantation of tumor cells. Another significant difference between the two groups was that the nodules were significantly larger in the sham-treated group than in the resveratrol-treated group. Moreover, tumors weighed 0.46 ± 0.07 g on average in the sham-treated group, while the average weight of the tumors was 0.10 ± 0.05 g in the resveratrol-treated group (Figure 6C), representing over 78% inhibition in tumor growth. Thus, resveratrol inhibits the growth and metastasis of MDA231 human breast cancer in a xenograft-bearing mouse model.

### 2.6. Resveratrol Does Not Cause Side Effects in Mice

To confirm whether resveratrol treatment induces side effects, we measured the weights of the mice every week after resveratrol treatment. There were no significant differences in body weights among the normal, resveratrol-treated, and sham-treated groups (*p* > 0.05, Figure 6D). Based on further analyses of liver and kidney function, we found that there was no significant difference in the levels of serum ALT (Figure 6E), Cr (Figure 6F), AST (Figure 6G), or BUN (Figure 6H) among the normal, sham-treated, and resveratrol-treated groups (*p* > 0.05). These results indicate that resveratrol can be used safely in mice.

## 3. Discussion

Resveratrol is a natural polyphenol compound that belongs to the stilbene class [22,23,24,25]. Studies show that resveratrol has antioxidant, anti-inflammatory, and anti-tumor effects [26]. Because of the pharmacological effects and safety of resveratrol, it has been used widely as a dietary supplement, food additive, and natural health food [27]. Previous studies reported that the effective dose of resveratrol in vivo is 100 mg/kg [28]. Our study found that resveratrol inhibits tumor growth and spontaneous lung metastasis of breast cancer (Figure 6A–C) without marked changes in liver or kidney function or body weight in a mouse MDA231 human breast cancer xenograft model (Figure 6D–H). Other studies have confirmed that resveratrol can reduced the tumor growth of MDA-MB-231 cells in athymic nude mice, which is consistent with the results of this study [17]. In addition, resveratrol effectively reduced breast cancer cell migration (Figure 2). These findings suggest that resveratrol has great potential to affect breast cancer metastasis. However, Castillo-Pichardo et al. found that resveratrol promoted the growth and metastasis of xenograft breast tumors in SCDI mice [29]. This is contrary to our findings and the reasons need to be further studied.

Transcription factors, growth factors, inflammatory cytokines, chemotactic factors, and other proteins participate in the EMT process [30]. EMT plays an important role in the processes of invasion and metastasis in cancer [31]. EMT is usually characterized by a reduction in the expression of epithelial markers such as E-cadherin and an increase in the expression of mesenchymal markers such as vimentin, fibronectin, and α-SMA [32]. Studies have confirmed that TGF-β1 can induce EMT.

A previous study has shown that DNA hypermethylation can silence the expression of E-cadherin. Partial or complete methylated cells lack the expression of E-cadherin [33]. However, in addition to MDA231 cells, the expression of E-cadherin in tumor cells after demethylated drug treatment was increased [34]. As a member of TNBC, it is necessary to search for drugs that can inhibit the metastasis of MDA231 cells. It also has been reported that the resveratrol suppresses TGFβ-induced EMT in a Smad-dependent manner in MCF7 cells [35]. In the present study, we found that the phenotype of MDA231 breast cancer cells treated with 5 ng/mL TGF-β1 for 24 h underwent an interstitial change (Figure 3A). Additionally, the expression level of E-cadherin was decreased, and the expression levels of vimentin, fibronectin, Snail1, Slug, and α-SMA were increased. Moreover, this phenomenon was reversed by treatment with 12.5, 25, or 50 μM resveratrol for 24 h in a concentration-dependent manner (Figure 3A,C). The relevant EMT markers were also verified by immunofluorescence assay. Resveratrol incubation increased the expression of E-cadherin on the cell membrane and decreased the expression of vimentin in the cytoplasm (Figure 3D). These studies indicated that resveratrol can reverse TGF-β1-induced EMT in MDA231 cells.

The expression and secretion of several extracellular matrix-degrading proteolytic enzymes play important roles in the metastasis process. MMPs are the main proteases involved in tumor cell migration, proliferation, invasion, and metastasis [36]. Among them, MMP-2 and MMP-9 are key enzymes that participate in the metastatic process [37]. In addition, MMP-2 and MMP-9 are downstream target proteins regulated by TGF-β [38]. In the present study, resveratrol significantly inhibited the expression and secretion of MMP-2 and MMP-9 in breast cancer cells (Figure 5B), which indicated that resveratrol inhibited the migration of breast cancer cells through decreasing the expression and secretion of MMP-2 and MMP-9 in breast cancer cells.

Studies have indicated that it is possible to prevent cancer cell proliferation, invasion, and metastasis by inhibiting the PI3K/AKT pathway. Moreover, previous research has found that a PI3K/AKT inhibitor reversed EMT [39]. The PI3K/AKT signaling pathway is involved in the EMT process in mammary epithelial cells via mediating the TGF-β signaling pathway. TGF-β1 phosphorylates AKT/PKB and activates the AKT signaling pathway. When the PI3K/AKT signaling pathway was inhibited, TGF-β-induced breast cancer cell migration was inhibited [40]. In this study, the PI3K inhibitor LY290042 combined with TGF-β1 reduced cell migration and restored the epithelial phenotype of MDA231 cells. Furthermore, LY290042 combined with TGF-β1 increased the expression of E-cadherin and decreased the expression of vimentin, fibronectin, Snail1, Slug, and α-SMA (Figure 4A–C) compared with TGF-β1 treatment alone. Resveratrol combined with TGF-β1 produced the same effect as the inhibitor (Figure 3C). Resveratrol decreased the expression levels of P-PI3K and P-AKT and inhibited TGF-β1-induced cell migration (Figure 3B and Figure 5A). These results suggest that resveratrol reverses TGF-β1-induced EMT, possibly through the PI3K/AKT signaling pathway.

The transcription factors Snail1 and Slug play a central role in EMT. TGF-β can upregulate HMGA2 expression through a Smad3/Smad4-dependent pathway and the abnormal expression of HMGA2 can induce Snail1/2 expression, thereby participating in the regulation of the EMT process [41]. Our results showed that the expression levels of Smad3, P-Smad3, Slug, and Snail1 were decreased by resveratrol (Figure 3C and Figure 5A), which may be related to the role of resveratrol in reversing TGF-β1-induced EMT.

The TGF-β signaling pathway is involved mainly in regulating the EMT process through Smad-dependent and Smad-independent signaling pathways [42]. The Smad2 and Smad3 proteins play a significant role in the process of breast cancer EMT, which is regulated by TGF-β. The overexpression of both Smad2 and Smad3 can induce EMT in breast epithelial cells [43]. Dzwonek proposed that Smad3 is more important than Smad2 in TGF-β1-induced EMT [44]. In the present study, the TGF-β 1 receptor subtype ALK4/5/7 inhibitor SB431542 combined with TGF-β1 reduced the number of migrated cells and induced the reappearance of the epithelial phenotype in MDA31 cells. Furthermore, SB431542 combined with TGF-β1 reversed the TGF-β1-induced expression of the EMT markers E-cadherin, vimentin, fibronectin, Snail1, Slug, and α-SMA (Figure 4A–C). Additionally, resveratrol decreased the expression levels of Smad2, Smad3, P-Smad2, and P-Smad3, which were increased by TGF-β1 (Figure 5A). These results indicate that resveratrol reverses TGF-β1-induced EMT, which is closely related to Smad signaling.

## 4. Materials and Methods

### 4.1. Materials

Resveratrol was purchased from Shanghai Standardization for the Traditional Research Center and dissolved in dimethyl sulfoxide (DMSO). Human TGF-beta1 was obtained from PeproTech (USA). 3-(4,5-dimethylthiazol-2-yl)-5-(3-carboxymethoxyphenyl)-2-(4-sulfophenyl)-2H-tetrazolium, inner salt (MTS) was purchased from Promega (Madison, WI, USA). QuicBlock^TM^ Blocking Buffer for Immunol Staining, Antifade Mounting Medium with DAPI, QuicBlock^TM^ Secondary Antibody Dilution Buffer for Immunofluorescence and QuicBlock^TM^ Primary Antibody Dilution Buffer for Immunol Staining were purchased from Beyotime (Shanghai, China). Antibodies against MMP-2, MMP-9, α-SMA, Smad2, Smad3, P-Smad2, P-Smad3, Snail1, and Slug were obtained from Santa Cruz Biotechnology (Santa Cruz, CA, USA). Fibronectin, P-PI3K, PI3K, P-AKT, AKT, E-cadherin, and vimentin antibodies were purchased from Cell Signaling Technology (Boston, MA, USA). LY290042, SB431542, SP600125, PDTC, and PD98059 were purchased from Selleck Chemical (Houston, TX, USA). PDTC and PD98059 were purchased from MedChemExpresss (New Jersey, NJ, USA). IRDye^TM^ fluorescence antibodies were obtained from Li-Cor Bioscience (Lincoln, NE, USA).

### 4.2. Cell Culture

MDA231 cells were purchased from the American Type Culture Collection (Manassas, VA, USA). MDA453, MDA436, and BT549 cells were purchased from Fu Heng Biotechnology Co. Ltd. (Shanghai, China). MDA231 and BT549 cells were cultured in DMEM with 10% calf serum, 100 μg/mL streptomycin, and 100 U/mL penicillin at 37 °C with 5% CO_2_ in a humidified atmosphere. MDA453 and MDA436 cells were cultured in L15 culture medium with 2 mM glutamine, 0.01 mg/mL insulin, 10% calf serum, 100 μg/mL streptomycin, and 100 U/mL penicillin at 37 °C with no CO_2_ in a humidified atmosphere.

### 4.3. Cell Viability Assay

Cell viability was measured by the MTS assay. First, 5 × 10^4^ cells were added to 96-well culture plates and incubated overnight. Then, the cells were treated with 0, 12.5, 25, 50, or 100 μM resveratrol for 24, 48, or 72 h, and 20 μL of MTS was then added and incubated at 37 °C for 4 h. An ELISA plate reader (Baxter, VT, USA) was used to measure the optical density (OD) at 490 nm.

### 4.4. Migration Assays

Transwell plates (Corning, NY, USA) were used to analyze cell migration ability. A total of 1 × 10^5^ cells were seeded on the Transwell chamber and 200 μL of culture medium without serum was added. The lower compartment was filled with 10% FCS and different concentrations of resveratrol, 5 ng/mL TGF-β1, 10 μM LY290042, or 10 μM SB431542. After 24 h, the upper chambers with the residual cells were removed and the cells under the surface were stained with 0.5% crystal violet for 10 min and then fixed with 70% ethanol. Furthermore, six fields of vision (×100 magnification) were selected randomly and used to observe cell migration.

### 4.5. Effects of Resveratrol on Breast Cancer MDA231 Xenografts

Six-week-old female nude mice were fed at the Laboratory Animal Center at Shanghai University of Traditional Chinese Medicine and housed under pathogen-free conditions during the experimental period. A total of 3 × 10^6^ MDA231 cells, which were suspended in Matrigel, were transplanted into the mouse mammary fat pads. Then, 24 h after tumor cell inoculation, the mice were injected once every two days with 100 mg/kg resveratrol (resveratrol-treated group, *n* = 8) or 1% DMSO/10% Tween-80 in PBS (sham-treated group, *n* = 8) through the peritoneal cavity. Body and tumor weights of the mice were measured once per week. After eight weeks of tumor cell inoculation, the mice were sacrificed, and the lungs and tumors were removed. The lungs were sectioned and stained with haematoxylin and eosin (H&E). Pictures of representative tissues from each group were taken and used to count the metastatic nodules.

### 4.6. Liver and Kidney Function Tests

When the mice were sacrificed, one millilitre of blood was collected from the eyes and then quickly centrifuged for 10 min at 3000 rpm to obtain the serum. According to the manufacturer’s instructions, aminotransferase (ALT), aspartic transaminase (AST), creatinine (Cr), and urea nitrogen (BUN) colorimeter testing kits (Jiancheng Bioengineering Institute, Nanjing, China) were used to detect the levels of these parameters in the serum.

### 4.7. ELISA

After treatment with 0, 25, or 50 μM resveratrol for 24 h, cell culture supernatants were collected, and the expression of MMP-2 and MMP-9 was detected by ELISA using Human MMP-9 (R&D Systems, Minneapolis, MN, USA) and Human MMP-2 ELISA kits (RayBiotech, Norcross, GA, USA) according to the manufacturer’s instructions.

### 4.8. Immunofluorescence

The cells were cultured on confocal dishes treated with 5 ng/mL TGF-β1, 50 μM resveratrol, or TGF-β1+resveratrol for 24 h fixed with 4% paraformaldehyde for 30 min, and then stabilized in 0.5% Triton X-100 for 20 min. After three PBS washes and blocking with Quick BlockTM Blocking Buffer for Immunol Staining for 15 min, the cells were incubated with E-cadherin (1:200) and anti-vimentin (1:500) antibodies overnight at 4 °C. After washing, the cells were blocked from light, incubated with an anti-rabbit antibody for 60 min and counterstained with DAPI. The cells were observed and photographed with a confocal fluorescence microscope (LSM880, Zeiss, Jena, Germany).

### 4.9. Western Blot Analyses

Whole cell lysates were electrophoresed with 8% or 10% SDS-PAGE. The expression levels of MMP-2, MMP-9, fibronectin, α-SMA, PI3K, AKT, P-PI3K, P-AKT, Smad2, Smad3, P-Smad2, P-Smad3, E-cadherin, vimentin, Snail1, Slug, and GAPDH were detected by first incubating the samples with the respective primary antibodies (1:1000~5000) and then incubating the samples with IRDye^TM^ 700DX (red)-conjugated or IRDye^TM^ 800DX (green)-conjugated secondary antibodies (1:10,000~20,000) for visualization. Images were generated using an Odyssey Infrared Imaging System (Li-Cor Biosciences, NE, USA). Alpha Ease FC (FluorChem FC2) software was used for the quantitative analysis of the Western blots. GAPDH expression was used as an internal reference to standardize the relative expression of the proteins.

### 4.10. Statistical Analyses

All data are displayed as the mean ± SD of at least three independent experiments. Student’s *t*-test, one-way ANOVA, and multi-factor variance analysis were used for statistical data evaluations, and *p* < 0.05 was considered significant.

### 4.11. Ethics Approval and Consent to Participate

All animal procedures were conducted in accordance with the guidelines of the National Institutes of Health and were approved by the Ethical Committee of the Shanghai University of Traditional Chinese Medicine (Approval ID: PZSHUTCM18101804).

## 5. Conclusions

In this study, we firstly found that resveratrol inhibited lung metastasis in xenograft-bearing mouse models of MDA231 human breast cancer. Resveratrol also inhibited the migration of MDA231 cells by downregulating the expression of MMP-2 and MMP-9. Furthermore, resveratrol inhibited TGF-β1-induced EMT in MDA231 cells by regulating the expression of PI3K/AKT, Smad, and MMP signaling molecules.

## Figures and Tables

**Figure 1 molecules-24-01131-f001:** Effect of resveratrol on breast cancer cell viability. The effect of resveratrol on the survival rate of MDA231, MDA453, MDA436, and BT549 cells was quantified by the 3-(4,5-dimethylthiazol-2-yl)-5-(3-carboxymethoxyphenyl)-2-(4-sulfophenyl)-2H-tetrazolium (MTS) method. Resveratrol was administered at concentrations of 0, 12.5, 25, 50, and 100 μM for 24, 48, or 72 h. The results are presented as the mean ± SD of three independent experiments; the SD is denoted by error bars. * *p* < 0.05, ** *p* < 0.01 vs. untreated cells.

**Figure 2 molecules-24-01131-f002:** Effect of resveratrol on the migration of MDA231, MDA453, MDA436, and BT549 human breast cancer cells. Transwell chambers were used to detect the ability of cells to migrate (×100 magnification). Cells were treated with control or 12.5 μM, 25 μM, or 50 μM resveratrol for 24 h. The percent cell migration is shown. The error bars represent three independent experiments and each experiment was repeated three times. * *p* < 0.05, ** *p* < 0.01 vs. untreated cells.

**Figure 3 molecules-24-01131-f003:** Effects of resveratrol on transforming growth factor (TGF) -β1-induced epithelial-mesenchymal transition (EMT) and cell migration. (**A**) Morphology of MDA231 cells treated with TGF-β1 and resveratrol. Images were captured by brightfield microscopy (200× magnification). MDA231 cells were untreated or treated with 5 ng/mL TGF-β1 or 50 μM resveratrol and 5 ng/mL TGF-β1 for 24 h. TGF-β1 induced morphological changes in mesenchymal cells in the MDA231 cell line: intercellular connections disappeared. However, this effect was reversed by resveratrol. (**B**) Transwell chambers were used to detect the ability of cell migration (×50 magnification). MDA231 cells were untreated or treated with 5 ng/mL TGF-β1, 12.5 μM resveratrol and 5 ng/mL TGF-β1, 25 μM resveratrol and 5 ng/mL TGF-β1, or 50 μM resveratrol and 5 ng/mL TGF-β1 for 24 h. The percent cell migration is shown. The error bars represent three independent experiments, and each experiment was repeated three times. * *p* < 0.05, ** *p* < 0.01 vs. cells treated with 5 ng/mL TGF-β1. # *p* < 0.05 vs. untreated cells. (**C**) EMT markers in MDA231 cells were detected by Western blot analysis. MDA231 cells were untreated or treated with TGF-β1 (5 ng/mL) and 12.5, 25, or 50 μM resveratrol for 24 h. The values are expressed as the mean ± SD. The experiments were repeated 3 times. # *p* < 0.05 vs. untreated cells; * *p* < 0.05 vs. cells treated with 5 ng/mL TGF-β1. (**D**) An immunofluorescence assay was used to detect EMT-associated proteins in MDA231 cells. MDA231 cells were untreated or treated with TGF-β1 (5 ng/mL) and/or 50 μM resveratrol for 24 h. E-cadherin and vimentin were stained red, and nuclei were stained blue with 4′,6-diamidino-2-phenylindole (DAPI). The images were captured at ×625 magnification.

**Figure 4 molecules-24-01131-f004:** TGF-β1 induced EMT in MDA231 cells through the PI3K/AKT and Smad pathways. (**A**) Cells were treated with or without the following for 24 h: 5 ng/mL TGF-β1; 1, LY290042 (PI3K inhibitor); 2, SB431542 (ALK4/5/7 inhibitor); 3, SP600125 (JNK inhibitor); 4, PDTC (NF-κB inhibitor); 5, PD98059 (ERK inhibitor). The proteins were evaluated by Western blot analysis. The values are expressed as the mean ± SD. The experiments were repeated three times. # *p* < 0.05 vs. untreated cells; * *p* < 0.05 vs. cells treated with 5 ng/mL TGF-β1. (**B**) Morphology of MDA231 cells treated with TGF-β1 and inhibitors. The images were captured by brightfield microscopy (200× magnification). Cells were treated with or without 5 ng/mL TGF-β1, TGF-β1+LY290042 or TGF-β1+SB431542 for 24 h. TGF-β1 induced morphological changes in mesenchymal cells in the MDA231 cell line: intercellular connections disappeared. However, this effect was reversed by the inhibitors. (**C**) Transwell chambers were used to detect the extent of cell migration (×100 magnification). MDA231 cells were untreated or treated with 5 ng/mL TGF-β1, 10 μM LY290042 and 5 ng/mL TGF-β1, or 10 μM SB431542 and 5 ng/mL TGF-β1 for 24 h. The percent cell migration is shown. The error bars represent three independent experiments, and each experiment was repeated three times. * *p* < 0.05 vs. cells treated with 5 ng/mL TGF-β1. # *p* < 0.05 vs. untreated cells.

**Figure 5 molecules-24-01131-f005:** The effect of resveratrol on TGF-β1-induced EMT through the PI3K/AKT and Smad pathways. (**A**) Cells were treated with or without 5 ng/mL TGF-β1 and 12.5, 25, or 50 μM resveratrol for 24 h. The protein expression levels were evaluated by Western blot analysis. The values are expressed as the mean ± SD. The experiments were repeated three times. # *p* < 0.05 vs. untreated cells; * *p* < 0.05 vs. cells treated with 5 ng/mL TGF-β1. (**B**) In MDA231 cells, resveratrol reduced the extracellular secretion of MMP-2 and MMP-9. MDA231 cells were treated with different concentrations of resveratrol for up to 24 h. We collected the cell culture medium from each treatment and analysed it with MMP-2 and MMP-9 ELISA kits according to the manufacturer’s instructions. Each experiment was repeated 3 times. ** *p* < 0.01, vs. untreated control.

**Figure 6 molecules-24-01131-f006:** Effects of resveratrol on tumor growth and lung metastasis in a MDA231 human breast cancer xenograft model. Mice were given 40 mg/kg resveratrol (i.p.) for 8 weeks (*n* = 8). (**A**) After removal, the lung tissues were fixed with Bouin’s solution for 24 h. Then, the metastatic nodules were counted under a dissecting microscope (×100 magnification). Typical histological findings in the lungs of saline- and resveratrol-treated mice are shown. (**B**) Quantification of pulmonary metastatic nodules. (**C**) Tumor weights of saline- and resveratrol-treated mice. (**D**–**H**) In treated mice, resveratrol did not induce significant toxicity. (**D**) For eight consecutive weeks, the sham- and resveratrol-treated mice were weighed weekly. ALT (**E**), Cr (**F**), AST (**G**), and BUN (**H**) levels in the blood of mice receiving sham or resveratrol treatment for eight weeks according to the respective colorimetric assay kits. The data are presented as the mean ± SD (*n* = 8). ** *p* < 0.01 vs. saline-treated group.

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
