# Peer review of "Resveratrol Inhibits the Migration and Metastasis of MDA-MB-231 Human Breast Cancer by Reversing TGF-β1-Induced Epithelial-Mesenchymal Transition"

_molecules, 2019, doi:10.3390/molecules24061131_

Round 1

Reviewer 1 Report

The focus of the study described in the manuscript is to investigate the efficacy and mechanism of action of resveratrol, a naturally found antioxidant, in suppressing epithelial-mesenchymal transition and metastasis in human breast cancer cell lines. The authors use cell survival and transwell migration assays to demonstrate the anti-survival and anti-migratory effect of resveratrol on four different breast cancer cell lines including MDA231 cells. To demonstrate the ability of resveratrol in suppressing TGFβ1-induced EMT, the authors use morphological screens, immunoblotting and immunofluorescence analyses focusing only on the MDA-MB231 cell line. They observe that the antioxidant suppresses both the Smad and PI3K/Akt pathways. Finally, they used an orthotopic athymic nude mice in vivo model to indicate that the lung metastases were both reduced in number and weight.

Overall, the study is of interest for a broad range of audience given the high mortality associated with triple negative breast cancer and the limited therapeutic options available.  However, previous published reports that have investigated potential use of resveratrol for the management of breast cancer using in vitro and in vivo approaches (reviewed in Sinha et al, 2016) should be cited in the introduction or discussion (in all breast cancer subtypes). In particular, there are conflicting findings on the effect of resveratrol on MDA231 cells in vivo, which should be discussed in the current manuscript in the context of the results described here. Finally, resveratrol suppression of TGFβ-induced EMT in a Smad-dependent manner in MCF7 cells has been already described (Shi et. al. 2013) raising the question of the novelty of the current study as well as relevance for this agent for TNBC since MCF7 represent a luminal breast cancer cell line.

Other specific comments:

1.     There is debate in the literature on whether MDA-MB-231 cells are a good model of EMT, at least in the context of two-dimensional culture settings, as these cells are more mesenchymal in nature with E-cadherin promoter hypermethylation.  Thus, the rationale for using MDA-MB231 cells among the other ones used in the first two figures needs to be explained.

2.     TGFß treatment should be included in the experimental design of results shown in figures 1 and 2, since the focus of this paper on this pathway.

3.     Resveratrol only treatment should be included in Figure 3 and y.

4.     Figure 4 seems redundant as the effect of these inhibitors have already been heavily tested in multiple different studies and do not add any new knowledge. Also, the actual activation of the pathway needs to be shown in the form of the status of downstream phosphorylated proteins for each signalling axis.

5.     It is very difficult to decipher any nodules in Figures 6A and the image should be replaced with a color image with the proper staining.

6.     In line 304, the SB431542 is not a Smad inhibitor per se but a selective small molecule inhibitor of the ser/thr kinase moiety of the ALK4/5/7 subtypes of the TGFß type I receptor, which should be corrected. 

7.     In line 62, the reference [10] notation needs to be shifted to the end of the sentence.

8.     The methods section is very brief lacking among others a description of treatment conditions.

9.     The manuscript will benefit from a spelling and grammar check.

Author Response

Dear Reviewer,

Thank you very much for your letter about our manuscript title of Resveratrol Inhibits the Migration and Metastasis of MDA-MB-231 Human Breast Cancer by Reversing TGF-β1-Induced Epithelial-Mesenchymal Transition. We have carefully revised and supplemented the manuscript. The following is our response to each of your comments.

Point 1: There is debate in the literature on whether MDA-MB-231 cells are a good model of EMT, at least in the context of two-dimensional culture settings, as these cells are more mesenchymal in nature with E-cadherin promoter hypermethylation.  Thus, the rationale for using MDA-MB231 cells among the other ones used in the first two figures needs to be explained.

Response 1: Thanks for the suggestion. In order to investigate the inhibitory effect of resveratrol on migration and metastasis of human breast cancer, we used MDA-MB-231 cells in the first two figures. The reason for this is that MDA-MB-231 cell is a type of triple negative breast cancer cell with high metastasis, and MDA-MB-231 cell has been reported as a TGF-β-induced EMT model [1-3]. 

[1] Sengupta, S.; Jana, S.; Biswas, S.; Mandal, P.K. Bhattacharyya A.Cooperative involvement of NFAT and SnoN mediates transforming growth factor-β (TGF-β) induced EMT in metastatic breast cancer (MDA-MB 231) cells. Clin Exp Metastasis.2013, 30:1019-1031.

[2] Park, HJ.; Kim, MK.; Choi, K.S.; Jeong, J.W.; Bae, S.K.; Kim, H.J.; Bae, M.K. Neuromedin B receptor antagonism inhibits migration, invasion, and epithelial-mesenchymal transition of breast cancer cells.Int J Oncol. 2016, 49:934-342.

[3] Shao, S.; Zhao, X.; Zhang, X.; Luo, M.; Zuo, X.; Huang, S.; Wang, Y.; Gu, S.; Zhao, X.Notch1 signaling regulates the epithelial-mesenchymal transition and invasion of breast cancer in a Slug-dependent manner. Mol Cancer. 2015, 14:28.

Point 2: TGF-ß treatment should be included in the experimental design of results shown in figures 1 and 2, since the focus of this paper on this pathway.

Response 2: Thanks for the comments. We did not include TGF-β treatment in figures 1 and 2 for the following reasons. TGF-β treatment was not included in the experimental design of figure 1, which aim is to observe an appropriate action time and concentration of resveratrol, providing reference for subsequent experiments. TGF-β treatment was not included in the experimental design shown in figure 2 in order to observe the inhibitory effect of resveratrol on the migration of tri-negative breast cancer cell lines. In figure 2, we observed that resveratrol 12.5, 25 and 50 μM can effectively inhibit the migration of MDA-MB-231 cells, so we selected MDA-MB-231 cells for subsequent experiments. For the focus of this paper on the pathway, TGF-ß treatment has been included in the experimental design of results shown in figures 3.

Point 3: Resveratrol only treatment should be included in Figure 3.

Response 3: Figure 3 is to observe the effects of resveratrol on TGF-β1-induced EMT and cell migrationthe. We aimed to verify the effect of resveratrol on the reversal of EMT model, so we did not include the resveratrol only treatment. Your suggestion is also very pertinent. Actually if resveratrol only treatment be included in Figure 3, the result may be more rigorous and intuitive. We will add the design of this experiment in further study.

Point 4:  Figure 4 seems redundant as the effect of these inhibitors have already been heavily tested in multiple different studies and do not add any new knowledge. Also, the actual activation of the pathway needs to be shown in the form of the status of downstream phosphorylated proteins for each signalling axis.

Response 4: Thanks for the suggestion. In figure 4 A, we have used 5 inhibitors that closely related to the TGF-β induced EMT pathway, and found that LY290042 and SB431542 may play a role in the EMT model of MDA-MB-231 cells. Figure 5 continues to observe the expression of PI3K/AKT and other related proteins. Therefore, we believe that retaining figure 4 may be more helpful to focus which signal pathways in the subsequent experiments and make the article more complete.

Point 5: It is very difficult to decipher any nodules in Figures 6A and the image should be replaced with a color image with the proper staining.

Response 5: According to the suggestion, we have replaced figure 6A with a color image with the proper staining.

Point 6: In line 304, the SB431542 is not a Smad inhibitor per se but a selective small molecule inhibitor of the ser/thr kinase moiety of the ALK4/5/7 subtypes of the TGFß type I receptor, which should be corrected.

Response 6: Thanks for the suggestion. This is indeed a mistake, and we have corrected it.

Point 7:  In line 62, the reference [10] notation needs to be shifted to the end of the sentence.

Response 7: Thank you very much for pointing out the inadequacies in line 62. We have shifted the [10] notation to the end of the sentence.

Point 8:  The methods section is very brief lacking among others a description of treatment conditions.

Response 8: According to the suggestion, we have added a description of treatment conditions in the Materials and methods section.

Point 9:  The manuscript will benefit from a spelling and grammar check.

Response 9: We have sent the manuscript to a language editing company to improve the article for language and style. We have attached the language editing certificate as a supplementary material.

Reviewer 2 Report

Submitted for review article entitled “  Resveratrol Inhibits the Migration and Metastasis of  MDA-MB-231 Human Breast Cancer by Reversing  TGF-β1-Induced Epithelial-Mesenchymal Transition.” is an original paper.

The authors undertook important topic which describes inhibition the migration and metastasis of  MDA-MB-231 Human Breast Cancer by Reversing  TGF-β1-Induced Epithelial-Mesenchymal Transition by resveratrol. Although resveratrol has already been strongly described in the literature,  new properties are still attributed to him. The methodology is good performed and reasonably clear. Furthermore, the statistical methodology is appropriate. The interpretation of the results is clearly presented and it is adequately supported by the evidence adduced but please explain why the authors chose MDA 231 for further experiments although cytotoxicity is similar in the MDA 453 line?  All the tables and figures are adequate and necessary. Discussion is clear and represented by a good choice of literature but unfortunately not the newest one.

Despite interesting presentation on the topic I have a few minor comments which are submitted below:

Minor comments:- in Figure 1 in the MDA 231 cell line is the number 0 should be control?

Authors should polish up their English and correct some tiny grammar mistakes. In line 80 is result should be results.

- the authors should explain abbreviations of all cell lines used in the study? MDA231, MDA453, MDA436 or BT549?

Final comments:

Article presented current problem, is well written and all methods were enough detailed. In my opinion article after minor modification should be considered for publication in Molecules.

Author Response

Dear Reviewer,

Thank you very much for your letter about our manuscript title of Resveratrol Inhibits the Migration and Metastasis of MDA-MB-231 Human Breast Cancer by Reversing TGF-β1-Induced Epithelial-Mesenchymal Transition. We have carefully revised and supplemented the manuscript. The following is our response to each of your comments.

Point 1: Please explain why the authors chose MDA 231 for further experiments although cytotoxicity is similar in the MDA 453 line?

Response 1: Thank you very much for your comments on this manuscript. In this study, all the mechanism studies were based on MDA-MB-231 cells. The reason is that  in the detection of cell migration ability, resveratrol did not significantly inhibit the migration ability of MDA-MB-453 cells. Therefore, only MDA231 cells were selected in this study to study the mechanism.

Point 2: In Figure 1 in the MDA 231 cell line is the number 0 should be control?

Response 2: Thank you very much for pointing out the inadequacies in Figure 1. We have changed "0" to "control".

Point 3: The authors should explain abbreviations of all cell lines used in the study? MDA231, MDA453, MDA436 or BT549?

Response 3: We have added abbreviations of all cell lines in the manuscript, and also added abbreviations in the LIST OF ABBREVIATIONS. Thank you very much for your comments.

Round 2

Reviewer 1 Report

The revised manuscript adequately deals with some of the previous comments, eg reference to previous studies on use of resveratrol in the management of breast cancer (Sinha et al, 2016).  However, the revised manuscript does not address other comments as stated below.  

1. The impact of a report showing that resveratrol suppresses TGFβ-induced EMT in a Smad-dependent manner in MCF7 cells (Shi et. al. 2013) on the novelty and relevance TNBC treatment by this reagent described in the current study should be addressed.

2. All three references suggesting the suitability of the MDA-MB-231 cell line as a good model of EMT should be included in the revised manuscript. The authors should also discuss this point in the context of other studies suggesting lack or low expression of E-cadherin due largely to promoter hypermethylation (Graff et.al. 1995, Lombaerts et. al. 2006). This information can be included in the introduction or discussion as part of the rationale for focusing on MDA-MB231 cells among the other cell lines used in the first two figures for the remainder of the study.

3.  TGFß treatment should be included in the experimental design of results shown in figures 1 and since the focus of this paper on this pathway.  This would allow to understand whether TGFß is involved in the survival and migratory nature of the different cell lines used.  The authors should consider using MCF7 cells in their studies.

4.  Resveratrol only treatment should be included in Figure 3.  The rationale provided for exclusion is weak.

5. In Figure 4, the actual activation of the pathway should be shown in the form of the status of downstream phosphorylated proteins for each signalling axis.

6. It is still difficult to decipher any nodules in the new Figures 6A.

Author Response

Dear Reviewer,

Thank you very much for your letter about our manuscript title of Resveratrol Inhibits the Migration and Metastasis of MDA-MB-231 Human Breast Cancer by Reversing TGF-β1-Induced Epithelial-Mesenchymal Transition. We have carefully revised and supplemented the manuscript. The following is our response to each of your comments.

Point 1:The impact of a report showing that resveratrol suppresses TGFβ-induced EMT in a Smad-dependent manner in MCF7 cells (Shi et. al. 2013) on the novelty and relevance TNBC treatment by this reagent described in the current study should be addressed.

Response 1: Thank you very much for the suggestion. Regarding the novelty and relevance of resveratrol in the treatment of TNBC in this study, we have added some treatment in "Conclusion".

Point 2:All three references suggesting the suitability of the MDA-MB-231 cell line as a good model of EMT should be included in the revised manuscript. The authors should also discuss this point in the context of other studies suggesting lack or low expression of E-cadherin due largely to promoter hypermethylation (Graff et.al. 1995, Lombaerts et. al. 2006). This information can be included in the introduction or discussion as part of the rationale for focusing on MDA-MB231 cells among the other cell lines used in the first two figures for the remainder of the study.

Response 2: Thanks for the comments. We have carefully studied the two literatures you provided and added the information of "lack or low expression of E-cadherin due largely to promoter hypermethylation" to the "Discussion".

Point 3:  TGFß treatment should be included in the experimental design of results shown in figures 1 and since the focus of this paper on this pathway.  This would allow to understand whether TGFß is involved in the survival and migratory nature of the different cell lines used.  The authors should consider using MCF7 cells in their studies.

Point 4:  Resveratrol only treatment should be included in Figure 3.  The rationale provided for exclusion is weak.

Point 5: In Figure 4, the actual activation of the pathway should be shown in the form of the status of downstream phosphorylated proteins for each signalling axis.

Point 6:It is still difficult to decipher any nodules in the new Figures 6A.

Response 3-6:

Thanks for the suggestion. Your comments are very pertinent and rigorous.Indeed, your comments will improve the quality of this manuscript. However, we feel very sorry, because now we do not have the time and conditions to conduct a large number of experiments to supplement. If conditions permit in the future , we will carefully consider supplementary experiments according to Suggestions of reviewers and continue to carry out further research.